# The Moderating Role of Team Conflict on Teams of Nursing Students

**DOI:** 10.3390/ijerph19074152

**Published:** 2022-03-31

**Authors:** Hsing-Yuan Liu

**Affiliations:** 1Department of Nursing, Chang Gung University of Science and Technology, No. 261, Wunhua 1st Rd., Gueishan Township, Taoyuan City 33303, Taiwan; hyliu@mail.cgust.edu.tw; 2Linkou Chang Gung Memorial Hospital, No. 5, Fuxing St., Guishan District, Taoyuan City 33305, Taiwan

**Keywords:** creativity, inter-professional education, team swift trust, team conflict, nursing students, nursing educators

## Abstract

Inter-professional education has become a widespread trend in healthcare education around the world. This study examined whether conflict moderated the correlation between swift trust and creativity for nursing students on teams in inter-professional education courses in Taiwan. A cross-sectional survey study with comparative, quantitative analysis was conducted to describe relationships between the studied variables. This study collected self-report data from 270 nursing students who attended interdisciplinary team-based capstone courses, and this study divided them into 54 teams. Each team consisted of five members. The study results showed cognition-based team swift trust had a positive correlation with team creativity. The negative association was revealed between relationship conflict and team creativity. Moderation models demonstrated that relationship conflict (95% C.I. [−0.70, −0.21]) negatively moderated the correlation between cognition-based swift trust and team creativity among nursing student teams. This research found that greater levels of cognition-based swift trust may enhance nursing students’ team creativity in inter-professional education courses. However, relationship conflicts may limit the positive outcomes of that association. Nursing educators should incorporate conflict management particularly aiming at relationship conflicts into their interdisciplinary nursing courses to support creative outcomes.

## 1. Introduction

Inter-professional education (IPE) is a common teaching model in health care education around the globe. According to the World Health Organization, IPE occurs “when students or members of two or more professions learn with, from and about each other” to advance interdisciplinary teamwork and improve healthcare quality [1]. Although education scholars sometimes disagree that the terms inter-professional and interdisciplinary are interchangeable, both IPE and interdisciplinary education programs involve diverse teams that must face the challenges of integrating and collaborating [2]. This study used IPE term throughout the study as our data were collected from students of two professions (nursing and design) to learn with, from and about each other to advance interdisciplinary teamwork and improve healthcare quality (develop healthcare-related products collaboratively).

Inter-professional or interdisciplinary courses tend to improve students’ collaborative skills and abilities [3]. Students who train in IPE courses have greater chances of becoming collaborative team members [4]. Several factors can increase the success of interdisciplinary collaboration, including swift trust, teamwork competency, and creativity. They are described as follows. In team settings, trust generates from a cognitive foundation oriented around equitability and skill, and an affective foundation that inspires a sense of emotional attachment to another person [5]. Accordingly, interpersonal trust on teams comprises two principal forms of trust: one based on cognition and the other based on affect [6]. The former employs an individual’s beliefs about their peers’ reliability and dependability, and the latter draws on their sense of interpersonal care and concern. In this study, cognition-based trust and affect-based trust were variables of swift trust on teams. A team was defined as a temporary team that formed by nursing and design students to collaboratively develop healthcare-related products in the capstone courses.

Swift trust, or trust developed quickly, helps diverse, temporary teams solve problems in short timeframes [5], such as quarter- or semester-long courses. Swift trust is also important in geographically distant teams for people who communicate electronically [2]. One study suggested a direct link between swift trust and creativity when team members rapidly organized a collaborative team environment; however, the study’s conceptual model lacked diversity of empirical testing [7]. Overall, research on swift trust in team-based IPE courses is lacking, especially regarding its relationship with team creativity.

Conflict that occurs in teams can comprise three types: task conflict, process conflict, and relationship conflict [8]. Task conflict can be observed when teammates have contradicting opinions and ideas about a task, and then they debate solutions to complete the task [9]. In process conflict, teammates disagree on who should contribute to task execution, and how and when it should be done [9]. Relationship conflict happens when team members perceive interpersonal mismatches and display irritation, hostility, or bitterness [9]. How these three types of conflict interact with swift trust and creativity regarding to nursing student teams in IPE courses is yet to be explored.

Generally, team trust and team creativity have been directly correlated [7,10,11,12]. However, the association between conflict and creativity in team settings has been less straightforward, and it varies with the type of conflict in question. Some team behavior scholars have reported that task conflict and creativity directly correlate to a degree, but then creativity falters when the level of task conflict is too great [13,14]. Other team researchers have reported that process conflict interferes with perceived creativity and performance [15], and relationship conflict negatively influences creativity [16,17]. Other evidence suggests that task conflict and creativity on teams are directly correlated [16,17,18,19].

Higher levels of team trust have been found to decrease team conflict [20]. Breaking conflict down into parts, constructive conflict, which often equates to task conflict, can strengthen team trust [21,22]. Additionally, as task (constructive) conflict increases, so does relationship (destructive) conflict; notably, team trust can moderate that effect, thus maintaining task (constructive) conflict and reducing relationship (destructive) conflict [23]. The existing literature does not address the relationships between swift trust, conflict, and creativity regarding nursing student teams in interdisciplinary settings.

This research not only sought to fill that information gap but provide nursing educators with guidelines for improving team creativity in their IPE courses.

The aforementioned studies proposed that a team’s swift trust and conflict may influence its creativity, and this study sought to understand the relationships between those variables regarding to nursing students in IPE courses in Taiwan. Furthermore, whether conflict can benefit or harm swift trust and its effect on collaboration and creativity in nursing IPE programs is still unknown. The literature that identified team trust as having the potential to moderate the direct association of task conflict with relationship conflict inspired me to examine whether any of the three types of conflict could have a moderating effect on the correlation of swift trust with creativity on interdisciplinary teams. Therefore, this study explored: (1) the relationships between measures of swift trust, conflict, and creativity regarding to nursing students on teams in IPE courses, and (2) the potential for conflict to moderate any association cognition-based and affect-based swift trust may have with creativity in those teams. Before collecting data from nursing student members of IPE teams, these hypotheses guided this research (Figure 1).

**Hypothesis** **1** **(H1).**
*A direct relationship will exist between swift trust and creativity on the teams.*


**Hypothesis** **2** **(H2).**
*Task conflict will have a direct association with team creativity, whereas conflicts of both process and relationship will have indirect associations with team creativity.*


**Hypothesis** **3** **(H3).**
*Conflict variables will negatively moderate any relationships between swift trust variables and creativity on the teams.*


## 2. Materials and Methods

### 2.1. Research Design, Participants, and Procedure

This non-experimental research followed a cross-sectional design with quantitative analysis to describe comparators. Using G*Power, this study determined that the minimum sample size required was 52 [24]. To establish a 95% confidence level, this study set the power of test at 0.8 and set 3 predictor variables. This study assumed 20% of students would not respond, and thus calculated 63 as the minimum required sample. 

Inclusion criteria were nursing students attending either of two consecutive 18-week IPE-based capstone courses, and who signed informed consent forms and submitted completed surveys. One capstone course ended in January 2018, and the other ended in January 2019. During class in the last week of each course, the first author presented the nursing students with the study’s purpose and procedure and asked them if they would like to participate. The 18-week IPE capstone course designed to help nursing students develop patentable healthcare-related products for clinical applications. This IPE capstone course is a selective course and is taught by two interdisciplinary faculties from the schools of nursing and design. The course includes lectures and small group discussions. Student teams are required to develop an innovative product, give a mid-term presentation, and present the final product at the end of the semester. The material covered included patent searching, healthcare product development, and needs assessment. Students received instruction in the use of creative thinking tools such as brainstorming to stimulate divergent and convergent thinking skills. This capstone course has been demonstrated to increase undergraduate nursing students’ creativity [25]. Therefore, in this study, both nursing and design students were enrolled in the course and randomly grouped into interdisciplinary teams of 5 students to develop a novel healthcare product. The students were from two universities of science and technology in northern Taiwan. Nursing students were from the school of nursing at one university. Design students were from the department of design at a second university.

The capstone courses were created to teach nursing students to collaborate in interdisciplinary settings and invent healthcare products they could patent for clinical applications. The capstone course design has demonstrated that when students in healthcare education work with students outside their discipline, their creativity improves [26,27]. Both capstone courses included students from two universities of science and technology in the north part of Taiwan; the nursing students were enrolled in the nursing program at one of the universities, and non-nursing students who provided interdisciplinary were enrolled in the other university’s design program. 

The IPE teams consisted of 5 members, including three or four nursing students and one or two design students, all of whom were randomly assigned to their teams. The faculty was also interdisciplinary, consisting of five instructors from the nursing program and two from the design program who delivered lectures and led small group discussions. All students met with their teams in person three times during the course for four-hour workshops, but they typically interacted online using video conferencing and instant messages. To facilitate remote team interactions, the faculty showed the teams how to organize online discussions to exchange task information and homework briefs. 

As teams, the students had to design a healthcare product, such as an easy-to-use urination device for women or an automatized intravenous injection controller that capstone course teams had created in the past. Each team could design and invent different healthcare products as they wished. The teams had to present their progress midway through the semester and show their final creations to a panel of three experts at the course finale. The experts were a clinical nurse, a medical engineer, and an industrial designer who critiqued the students’ final products. On the last day of each course, nursing students who chose to participate in the study answered self-report questionnaires, and this study collected their packets when they were finished. The students were provided with written informed consent forms; those who signed them were given a coded packet containing appropriate paper-and-pencil questionnaires, which included the students’ age and gender in the capstone course.

### 2.2. Instruments

Perceptions of both cognition-based and affect-based swift trust can be measured with a Swift Trust scale developed for multinational teams of MBA students by Kanawattanachai and Yoo [28]. This study used a Taiwanese version of the scale, which was designed for Chinese populations [29]. Swift Trust scale had good validity and reliability [28,29]. The Taiwanese version of Swift Trust scale is a 10-item instrument that had five items for the cognition-based component and five items for the affect-based component. Each component included five statements designed to gauge the students’ sense of trust on their teams. This study scored them using a 5-point Likert scale, ranging from 1 (strongly disagree) to 5 (strongly agree). The total score for team swift trust was the mean of its components; higher total scores signified greater levels of team swift trust. In this study, the Cronbach’s alpha coefficients were 0.72 for cognition-based swift trust and 0.70 for affect-based swift trust, indicating both had acceptable reliability. This study used a factor analysis to confirm that validity was acceptable (χ^2^/df = 2.945, RMSEA = 0.084, NFI = 0.906, CFI = 0.921, IFI = 0.922).

This study measured team conflict by modifying the 9-item instrument from Jehn and Mannix [15] had created for conflicts related to tasks, processes, and relationships; each component had good validity and reliability. In this study’s instrument, three open-ended questions asked participants to rate the frequency of each type of conflict on their team. The participants scored their answers on a 5-point Likert scale. Each total component score was the mean of its items, with higher scores indicating more of that type of team conflict. The instrument was reliable: the Cronbach’s alpha coefficients for all conflict components spanned from 0.77 to 0.94. A factor analysis ascertained that the instrument was acceptably valid (χ^2^/df = 3.255, RMSEA = 0.090, NFI = 0.912, CFI = 0.937, IFI = 0.938).

To measure team creativity, this study modified existing valid and reliable instruments by Farh et al. [13] and Li et al. [30], the latter of which was developed to measure creativity for management teams in Chinese populations [30]. The instrument for this study included 10 items to help participants rate the creativity on their teams along a 5-point Likert scale. The total team creativity score was the mean of all items, with higher scores indicating greater perceived team creativity. The instrument in this study had satisfactory reliability, as confirmed by the Cronbach’s alpha 0.94. Factor analysis verified acceptable validity (χ^2^/df = 4.213, RMSEA = 0.098, NFI = 0.910, CFI = 0.931, IFI = 0.933).

### 2.3. Data Analysis

This study used SPSS version 20.0 (IBM Corp., Armonk, NY, USA) to analyze the data. This study used mean and standard deviation (SD) as descriptive statistics to summarize participants’ demographics and instrument scores. This study used Pearson’s correlation analysis to display how the students’ scores on each instrument related to each other [31]. Once those correlations were clear, this study then employed the PROCESS macro by SPSS [32] to run multiple regression analyses and identify whether the relevant conflict variables would moderate the correlation of cognition-based swift trust (the independent variable) with creativity (the dependent variable) among teams. The macro is based on ordinary least-squares regression and bootstrapping, which is a powerful approach for the assessment of indirect effects, as it is free from assumptions regarding the shape of the sampling distribution of the indirect effect and has better control of type I errors [32]. This study used bias-corrected bootstrapping with 5000 resamples to generate 95% confidence intervals, which are significant at *p* < 0.05 if they do not contain zero [32]. Before conducting a regression analysis for moderating effects, this study used mean centering (subtracting raw scores from the mean) to bypass multicollinearity [33,34]. In the moderation model, using the PROCESS macro, this study entered team creativity as the dependent variable, cognition-based swift trust was entered as the independent variable, and relationship conflict was entered as the moderator. 

### 2.4. Data Aggregation

With the teams as our units of analysis, this study aggregated individual participants’ responses to the instruments to calculate the team-level scores. To determine if aggregation was appropriate, this study calculated the inter-team-member agreement (rwg) for the individual scores [35]. The median value for the Rwg was 0.87 for swift trust, 0.78 for conflicts, and 0.95 for creativity, indicating the aggregations were valid and supported reporting our findings at the team level. According to [36], a median Rwg of each instrument’s scale equal to or exceeding 0.70 was considered appropriate for aggregating individual data and deriving the team-level scores. Therefore, mean scores represent team swift trust, team conflicts, and team creativity.

This study then calculated ICC (1) and ICC (2) for team swift trust, conflicts, and team creativity to assess the data aggregation reliability. The ICC (1) = 0.41, ICC (2) = 0.87 for team swift trust, ICC (1) = 0.49, ICC (2) = 0.93 for conflicts, and ICC (1) = 0.68, ICC (2) = 0.94 for team creativity. The intra-class correlation, or the intra-class correlation coefficient (ICC), is a descriptive statistic that can be used when quantitative measurements are made on units that are organized into groups. ICC (1) is an estimate of interrater reliability at the level of the individual rater. ICC (2) is used to measure the reliability for the aggregated level data. According to Myers [37], the aggregation analysis is appropriate when the ICC values greater than 0.25. Hence, the ICC values results also indicated our swift trust, conflicts, and creativity validated the aggregation of findings at the team level. Therefore, this study aggregated every team member’s response to generate total scores for each of the two teams [36].

### 2.5. Ethical Approval

The Institutional Review Board (IRB) of the hospital ethics committees approved this study prior to data collection. The first author described the study and its purpose to the nursing students and invited them to participate during the last week of their capstone courses. Students were informed of their right to withdraw their participation at any time, for any reason. All data were collected anonymously. Only packets containing a signed written informed consent form were included in our analysis. The director of the healthcare program checked for the presence of a signed consent form and removed it from the packet to ensure the participants’ confidentiality.

## 3. Results

### 3.1. Participant Characteristics

Of the 270 nursing students in our study, the majority of students were females (83%), and the mean age was 21.4 years (SD = 0.93). Students were divided into 54 teams. Each team consisted of five members. Table 1 shows the mean scores for items of cognitive-based and affect-based components of swift trust. Data aggregation resulted in mean scores of cognitive-based team swift trust of 3.61 (SD = 0.86) and affect-based team swift trust of 2.53 (SD = 0.34). 

Table 2 shows the mean scores for the items of the three components of conflicts. Mean scores for the components of conflicts were 2.90 for task conflict (SD = 0.52), 2.63 for process conflict (SD = 0.62), and 2.18 for relationship conflict (SD = 0.78). Table 2 shows the mean scores for the items of team creativity. The mean score for team creativity was 3.81 (SD = 0.34).

### 3.2. Correlations between Swift Trust, Conflict, and Creativity on Teams

According to Pearson’s correlation analysis (Table 2) of the aggregated teams, team creativity was positively correlated with cognition-based team swift trust (d = 0.4, *p* < 0.01); however, it was negatively correlated with relationship conflict (d = 0.4, *p* < 0.05). Table 2 also shows that cognition-based team swift trust was negatively correlated with all three team conflict components (d = 0.6 for task conflict, d = 0.5 for process conflict, d = 0.8 for relationship conflict, *p* < 0.01). Affect-based team swift trust were also negatively correlated with all three team conflict components (d = 0.3 for task conflict, d = 0.4 for relationship conflict, *p* < 0.01) except process conflict (d = 0.5, *p* = 0.41).

### 3.3. Moderation Effects

Creativity was significantly correlated with cognition-based swift trust and relationship conflict on teams, suggesting relationship conflict as a potential moderator of the association between cognition-based swift trust and creativity. The moderation analysis for cognition-based swift trust and relationship conflict is shown in Table 3. This study regressed the total team creativity score for all teams (*N* = 54) on relationship conflict and its interaction with the total cognition-based swift trust score. The interaction between the relationship conflict and the total cognition-based swift trust score was significant (β = −0.453, 95% C.I. [−0.702, −0.205], *p* < 0.001), indicating that relationship conflict moderated the correlation between the cognition-based swift trust and team creativity.

## 4. Discussion

This study investigated how variables of swift trust, conflict, and creativity related to each other regarding to nursing students on IPE teams in Taiwan. Following the studies of Liu [25] and Liu [38], this study aggregated individual participants’ responses to the instruments to calculate the team-level scores. According to Pirola-Merlo and Mann’s study [39], failure to account for aggregation across individuals can result in misleading empirical results. Therefore, team-level scores of swift trust, conflict, and creativity are important to determine by aggregation processes across team members. The significant direct relationship between creativity and cognition-based swift trust among the interdisciplinary teams partially supports the first hypothesis, which expected the same for both types of swift trust. In other words, cognition-based swift trust could be the predictor of team creativity. Given results of Berthold [9], this study also expected team swift trust to predict perceived team creativity. However, the short course duration may have given insufficient time for the students to build enough swift trust to predict team creativity. A longer period of time might have shown a different outcome as team members developed increased knowledge about each other [40]. Although the results aligned with studies detecting a direct association between swift trust and creativity on teams [7,10,11,12], those studies had assessed swift trust as one variable rather than as two separate components. Therefore, the results support findings that cognition-based trust plays a larger role than affect-based trust in forming and maintaining swift trust on short-term teams [5,28]. When teams collaborate over longer periods of time that allow team members to learn more about one another personally, affect-based trust may supersede cognition-based trust [40]. Longitudinal studies could help uncover whether long-term interdisciplinary teams exhibit differences in cognition- and affect-based swift trust. Longitudinal studies could also help determine whether the cognition- and affect-based components of team swift trust change on interdisciplinary teams that collaborate over greater lengths of time, and to what degree they affect perceived team creativity.

Our findings did not support the second hypothesis: they could be seen as a result of the fact that task conflict has a direct association with team creativity, whereas conflicts of both process and relationship have indirect associations with team creativity. Only the component of relationship conflict was found to negatively correlate with team creativity, which is consistent with others [1,16,17], suggesting that higher levels of relationship conflict discouraged team members from sharing information and knowledge with each other, thereby diminishing team creativity [16]. In order to avoid such an outcome, nursing educators could ensure that IPE teams employ strategies to manage conflict and reduce the detrimental impacts that relationship conflict could have on creativity [41,42]. Inconsistent with other team studies [16,17,18,19], this study did not find that task-related conflict correlated directly with creativity regarding to nursing students on IPE teams. Task conflicts on the interdisciplinary teams may have had a constructive effect that helped enhance creativity, perhaps because the team members had to have open discussions and debates to understand and work with the diversity of thoughts and ideas [16]. However, our findings are not in line with Lee et al.’s [16] study’s implication above. One possible explanation could be nursing students in IPE programs encounter more task conflict with design students as they need design students to help them complete task of developing healthcare-related products in IPE-based capstone courses. 

On the other hand, the results indicated that relationship conflict correlated indirectly with creativity on the teams. Consistent with others [1,16,17], the finding suggests that higher levels of relationship conflict discouraged team members from sharing information and knowledge with each other, thereby diminishing team creativity [16]. In order to avoid such an outcome, nursing educators could ensure that IPE teams employ strategies to manage conflict and reduce the detrimental impacts that relationship conflict could have on creativity [41,42].

Unlike team research that had found strong indirect correlations between process conflict and creativity [43], this study revealed an insignificant direct relationship between those variables. This discrepancy may indicate the mixed nature of findings on how process conflict affects positive team outcomes, including that process conflict can remind team members to seek help, clarify roles, allocate resources, and plan for effective time management [44]. Furthermore, process conflict may be more complex than this study had assumed; had this study divided process conflict into sub-components for logistical conflict and contribution conflict [25], the study may have achieved a clearer sense for potential nuance. Nonetheless, perhaps the process conflict that existed on the teams in this study prompted them to be more concerned about allocating time and resources in a way that supported creative outcomes.

Finally, the moderation analysis partially confirmed the third hypothesis. Whereas the study had expected all forms of conflict to negatively moderate the relationship between swift trust and creativity, task conflict and process conflict had no association with creativity, so this study excluded them from the moderation analysis. Then, relationship conflict had negative moderation effects on the correlation of cognition-based swift trust with creativity. The finding suggests that relationship conflict—one associated with higher creativity and the other associated with lower creativity—may have together blurred the direct association between cognition-based swift trust and creativity regarding to nursing students on teams in IPE settings. Although the students rated relationship conflict as lower than task conflict on their teams, the former may have limited their sense of cognition-based swift trust enough to drive the negative moderation effect the study found. In any case, nursing educators may facilitate team creativity for their nursing students attending interdisciplinary capstone courses in Taiwan by including conflict management strategies that de-emphasize conflicts of relationship [41,42].

### 4.1. Limitations

This cross-sectional, quantitative study included only nursing students from capstone courses following an IPE format in Taiwan, eliminating the possibility of finding causation or generalizing the findings to interdisciplinary teams elsewhere. Furthermore, this study measured swift trust, conflict, and creativity according to the nursing students’ subjective views, but did not obtain objective data that could uphold or disprove their validity. Finally, future research in this area could assess the data using before and after research design to evaluate whether nursing students have changed their attitudes of teamwork after attending the interdisciplinary courses and be more creative in practice after graduation. 

### 4.2. Implications for Nursing Education

Nursing educators can use this study to understand more about the dynamics of interdisciplinary teams and increase the creative success of their students. This study found that nursing students of IPE teams quickly built cognition-based trust, which can increase team creativity. With that information, nursing educators can choose to promote cognition-based trust in IPE courses from the start. This study also found that two types of conflict effected negative moderation on the direct correlation of cognition-based trust with creativity, suggesting that nursing educators should incorporate conflict management into their interdisciplinary nursing courses to support creative outcomes.

## 5. Conclusions

This study makes contributions to the literature on interaction behaviors and creativity of teams of nursing students in three ways. First, this study is the first to our knowledge to explore the relationships between interpersonal interactions that affect how collaborations of nursing students are in capstone courses with an IPE format. At the end of interdisciplinary capstone courses created to encourage nursing students to collaborate with non-nursing students to develop healthcare products, cognition-based swift trust had a direct association with creativity, but affect-based swift trust was unrelated. Second, our study expanded on these previous findings by assessing swift trust as two separate elements and conflicts as three components, rather than one. Thus, our findings provide additional support that cognitive-based swift trust plays a more important role than the affective element for the formation and maintenance of swift trust in temporary teams. Moderation analyses showed that relationship conflicts exerted negative moderation effects on the correlation of cognition-based swift trust with creativity. Third, our findings also provide additional support that relationship conflicts play more important roles than the process conflict component for the indirect effects on the relationship of cognition-based swift trust with creativity on the teams. This study can guide nursing educators to implement effective IPE in development effective IPE in nursing programs, particularly for capstone courses in Taiwan.

## Figures and Tables

**Figure 1 ijerph-19-04152-f001:**
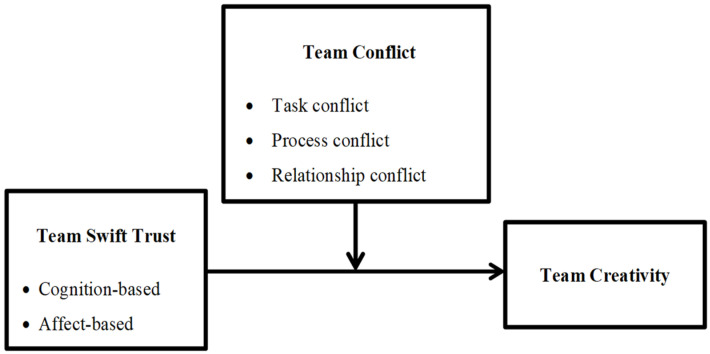
Hypothesized moderating effect model by team conflict variables.

**Table 1 ijerph-19-04152-t001:** Participants’ demographic characteristics and aggregation mean scores for teams (*N* = 54).

Variables	Mean	Std. Dev.	Min.	Max.	No. of Observation (*N* = 270)
Age (Year)	21.40	0.93	19	27	
Sex					
Male					45
Female					225
Team Creativity	3.81	0.34	2.67	4.44	
Team Swift Trust					
Cognition-based	3.61	0.86	1.40	4.80	
Affect-based	2.53	0.64	1.20	4.60	
Team Conflict					
Task Conflict	2.90	0.52	1.93	4.13	
Process Conflict	2.63	0.62	1.33	4.13	
Relationship Conflict	2.18	0.78	1.13	4.13	

Notes: Each team consists of 5 members, total observation = 54 × 5 = 270 (*N* = 54).

**Table 2 ijerph-19-04152-t002:** Pearson’s correlations of student teams’ (*N* = 54) scores on the team swift trust, team conflict, and team creativity instruments.

	Team Creativity	Team Swift Trust		Team Conflict		
		CB	AB	TC	PC	RC
Team Creativity						
Team Swift Trust						
Cognition-based (CB)	0.35 **					
Affect-based (AB)	0.06	0.50 **				
Team Conflict						
Task Conflict (TC)	−0.01	−0.59 **	−0.33 **			
Process Conflict (PC)	−0.05	−0.50 **	−0.12	0.80 **		
Relationship Conflict (RC)	−0.43 **	−0.81 *	−0.37 **	0.76 **	−0.43 **	

Notes: SE: standard error; Each team consists of 5 members. ** Significant at 0.01; * Significant at 0.05.

**Table 3 ijerph-19-04152-t003:** Regression analysis parameters examining whether the scores for relationship conflict (RC) for student teams (*N* = 54) moderate relationships between cognition-based swift trust subscale and team creativity scores.

Variables	β	SE	95% Confidence Interval	t	*p*-Value
Constant	−0.36	0.149	(−0.66 to −0.06)	−2.422	0.019
Relationship conflict (centered)	−0.61	0.199	(−1.01 to −0.21)	−3.055	<0.01
Cognition-based swift trust (centered)	0.02	0.195	(−0.37 to 0.41)	0.117	0.907
Relationship Conflict × Cognition-based swift trust	−0.45	0.124	(−0.70 to −0.21)	−3.662	<0.001

Notes: SE: standard error; Dependent variable: team creativity; Each team consists of 5 members.

## Data Availability

The data that support the findings of this study are available on request from the corresponding author, H.-Y.L.

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
