# Peer review of "The Moderating Role of Team Conflict on Teams of Nursing Students"

_ijerph, 2022, doi:10.3390/ijerph19074152_

Round 1
Reviewer 1 Report
Dear author,
Balancing between trust and conflicts is essential to the nursing arena. In my opinion, this manuscript is a bit similar to your other publications.
In addition, two minor comments: 1) please describe the ethical aspects of the methods.
2) I suggest providing the full names of your variables in Tables 1,2,3 rather than writing the acronyms.
I think that you could deepen the discussion about cognition-based swift trust.
Reviewer 2 Report
This article described if there is a direct connection between swift trust and creativity in interdisciplinary educational teams or/and how (or if) the latter has been influenced by team conflict.
Some general suggestions: you sometimes used “I” and sometimes wrote about “we”: please review this on the whole manuscript and decide which one to use.
It sounds a little bit strange to me, that you report about teams/IPE/interdisciplinarity and then you are alone as the author. Maybe it would help the readers if you explicitly state what your role in the faculty is? Were you involved in the training courses? If yes, please describe your role. If not, make it more clear how you were connected with the faculty.
I would suggest using also one other term more consistently opt possibly or for interdisciplinary or for IPE, or even only speak about teams.
Please rethink your title and make it fit better to your study.
In the introduction section, you started the article that IPE is a “common teaching model” and refer to the definition of WHO and that you used the term interdisciplinarity and IPE in an interchangeable manner in the first paragraph. It’s fancy to use both terms, but I would say in your article you report on some important elements/variables which can influence the potential of/can have an impact on the collaboration of teams with a clear task.
After this sentence for me it’s not enough to write the sentence ” …as follows” (line 39) trying to connect the content from before with different components of “trust” in “team settings”: I would say that both components of trust are important to be developed for working in a team/ team settings. (I would avoid using words like “stems” and “engenders”)
You could also add a sentence about how “team” is defined in the literature and/or how you defined/used it in the study.
After line 47 your introduction sounds more smooth and you write clearly about the different variables important for this study.
In the Methods part: in line 104 where you write about the inclusion criteria, I think it would further improve the quality of your reporting if you would describe the capstone courses a little bit more detailed: 18 weeks is a long period. What is the content of the courses? How are they distributed over the education period, how many hours do students work in the courses per week? Is it a compulsory course or can the students opt for it? Since when are they offered?
How were the teams formed?
In line 119 you write that the IPE teams consisted of 6-7 nursing students and 1-2 design students. For me, it is a gap between what you reported in the abstract/results (line 198) section (that all teams had 5 members). Could you fill the gap with information about team creation? Or were there other teams created from these teams?
In line 127 you stated that the teams had to design (as an outcome) a healthcare product: Could you please add a sentence in which you declare if every team had to design another product or if all had to design the same? Could a team also invent what it wanted?
In the next paragraph (131-134): Were there interferences between the team products and the questionnaires? How did the results from the questionnaires and the team products interfere?
(if applicable, report in the results section)
Could you please write how you did distribute/collect the questionnaires? (paper-based/online)
In line 137 (instruments) the word “translated” confused me, could it be that it was “available” in Chinese?
How did it “help” you to create a 10- item instrument?
In lines 143-144, 153-154, 161-162 you are reporting results obtained from the reliability analysis you did, or are these the values the different authors obtained in their original instruments?
Why you do not state in lines 145, 154, 162 the type of factor analysis (exploratory/confirmatory) you applied? Please also state the obtained values for validity and not only write that it was acceptable.
Regarding the Likert scales (lines 151, 159) for team conflict and –creativity: please state explicitly what the answer options were and if they were also ranging from 1-5 or 0-4?
In the data analysis paragraph (line 168-173) where you describe the variables you included in regression analysis: would you please state all independent variables and how you included them into your model (enter, forward, backward, stepwise).
In the data aggregation paragraph from lines 184-191: I would say it could enhance the reader's understanding if you would describe explicitly what the calculated ICC (1) and (2) stands for. Was ICC (1) from the course finished in 2018 and the (2) from that of 2019?
The 1st paragraph of the results section seems to be a part not overwritten from the author guidelines. Please instead use it to introduce the readers to this section results of your sample.
As described before please match the team's description with the research design section.
Table 1: I would suggest canceling the last column and adding the number 54 into the heading of the table. Please proof the sex indicated in the table. It’s the opposite of the one declared in the text before the table. Also add, as by the other tables, the notes in which you explain the abbreviations, or you could facilitate the reader indication of the full names?
Please control the spelling of the abbreviations in all tables, it seems that “RBST” should be “ABST”.
Please control the text under paragraph 3.2 from lines 210-215 and the results reported in table 2/ notes. Do you find also a mismatch? If yes, please correct. If not, please state them more clearly.
Why do you report under paragraph 3.3 the value of B and not the β value?
Please add the β value also as a column in table 3 and not only the SE β so that the reader can evaluate the real predictive potential of the variables.
In line 257 I think it would help the reader to rewrite in a discursive manner the 2nd hypothesis wording for your results such as that “they could be seen as a result…”
From my point of view you could in this paragraph always substitute the term IPE with “interdisciplinary”.
I feel that the word “dampened” in line 299 is confusing, could it be “blurred”. Please rethink the wording.
Under the limitations section: please add a phrase that your study design was another limitation. Find a better word for “causation” in line 310, because this is not possible to find a “cause” with a cross-sectional design.
I would change the sentence in line 313- on future research: cancel “more objective data… grades” but integrate data from other perspectives to have a more comprehensive picture. Another possibility would be a “before and after” design, assessing the variables before attending the course and after it, or evaluating how/if the interdisciplinary course changed the attitudes of the nurses (after graduation) are they more creative in practice?
Under “conclusions” you write that this study makes an “important” contribution. I would suggest letting this word away. In line 327 after “first”, I would suggest adding “ to our/my knowledge” ….
Please make the sentence in lines 328-329 better fitting to your findings. (you didn’t measure the success of students, didn’t you?).
In the last sentence please change the words “ to implement” into “in developing” effective ….It is more related to your findings.
Reviewer 3 Report
Dear Authors:
Congratulations on the work done. It is a correct and well-developed manuscript, both scientifically and formally.
In spite of this, some improvements can be made, for example, the addition of the effect size of some of the calculated statistics.
In addition, any abbreviations used in the tables should be described at the foot of the table (so that each figure is individually intelligible by itself).
This study explored the relationships between measures of rapid trust, conflict, and creativity with respect to nursing students in teams in interprofessional education courses, and the potential for conflict to moderate any cognition-based associations and affect-based rapid trust may have with creativity in those teams. Personally, I believe the authors have successfully developed the methodology to achieve these goals.
The research topic is relevant and of impact for professionals because it addresses practical methodological aspects of teaching in Health Sciences that we, as teachers, encounter on a daily basis.
This manuscript provides new knowledge in general for the teaching of university careers related to Health Sciences and, especially, for nursing.
I have not found similar research among the existing scientific literature.
This research found that greater levels of cognition-based trust may enhance nursing students’ team creativity in inter-professional education courses. However, relationship conflicts may limit the positive outcomes of that association. Nursing educators should incorporate conflict management particularly aiming at relationship conflicts into their interdisciplinary nursing courses to support creative outcomes. This should give health science teachers food for thought and should be taken into account in their classroom dynamics. Education in values and personal and social skills are fundamental in any profession, but this is an added value of incalculable value in Health Sciences.
The Conclusions and References are adequate and congruent with the research object analyzed.
Again, congratulations on the work done.
Kind regards.
Round 2
Reviewer 2 Report
Very well improved manuscript!